# Noncoding RNAs and Liquid Biopsy in Lung Cancer: A Literature Review

**DOI:** 10.3390/diagnostics9040216

**Published:** 2019-12-09

**Authors:** Antonia Haranguș, Ioana Berindan-Neagoe, Doina Adina Todea, Ioan Șimon, Mărioara Șimon

**Affiliations:** 1Research Center for Functional Genomics, Biomedicine and Translational Medicine, Iuliu Hatieganu University of Medicine and Pharmacy, 23 Marinescu Street, 400337 Cluj-Napoca, Romania; Antonia.harangus@yahoo.com (A.H.); ioananeagoe29@gmail.com (I.B.-N.); 2Leon Daniello Pneumophysiology Clinical Hospital, 40037 Cluj-Napoca, Romania; dtodea@umfcluj.ro (D.A.T.); simonmariaro@gmail.com (M.Ș.); 3MEDFUTURE-Research Center for Advanced Medicine, Iuliu Hatieganu University of Medicine and Pharmacy, 23 Marinescu Street, 400337 Cluj-Napoca, Romania; 4Department of Functional Genomics and Experimental Pathology, The Oncology Institute Prof. Dr. Ion Chiricuta, 34-36 Republicii Street, 400015 Cluj-Napoca, Romania; 5Departament of Pulmonology, Iuliu Hațieganu University of Medicine and Pharmacy, 40037 Cluj Napoca, Romania; 6Departament of Surgery, Iuliu Hațieganu University of Medicine and Pharmacy, 40037 Cluj Napoca, Romania

**Keywords:** lung cancer, noncoding RNA, liquid biopsy

## Abstract

Lung cancer represents a genetically heterogeneous disease with low survival rates. Recent data have evidenced key roles of noncoding RNAs in lung cancer initiation and progression. These functional RNA molecules that can act as both oncogenes and tumor suppressors may become future biomarkers and more efficient therapeutic targets. In the precision medicine era, circulating nucleic acids have the potential to reshape the management and prognosis of cancer patients. Detecting genomic alterations and level variations of circulating nucleic acids in liquid biopsy samples represents a noninvasive method for portraying tumor burden. Research is currently trying to validate the potential role of liquid biopsy in lung cancer screening, prognosis, monitoring of disease progression, and treatment response. However, this method requires complex detection assays, and implementation of plasma genotyping in clinical practice continues to be hindered by discrepancies that arise when compared to tissue genotyping. Understanding the genomic landscape of lung cancer is essential in order to provide useful and innovative research in the age of patient-tailored therapy. In this landscape, the noncoding RNAs play a crucial role due to their target genes that dramatically influence the tumor microenvironment and the response to therapy. This article addresses present and future possible roles of liquid biopsy in lung cancer. It also discusses how the complex role of noncoding RNAs in lung tumorigenesis could influence the management of this pathology.

## 1. Introduction

Lung cancer represents the most frequently diagnosed and the leading cause of cancer-related death in males on a global level. Among females, it follows breast and colorectal cancer-related death, standing at third place; it accounts for 11.6% of the total cases of cancer and 18.4% of the total cancer deaths [1]. Due to late diagnosis, together with the lack of curative therapies in an advanced disease stage, these patients have a low long-term survival. 

A lot of research has been undertaken to establish the role of genetics and genomics in the pathogenesis and progression of lung cancer since decisive DNA sequencing methods were popularized almost half a century ago [2]. The Human Genome Project and cancer genomic research unlocked the possibility to further understand the somatic modifications in cancer that could be used as a tool in prevention, early diagnosis, novel treatments, and resistance monitoring [3]. The challenges in the study of lung cancer genomics appear mainly as a result of intratumoral heterogeneity, technical limitations, costs, clinical translation, but also a lack of the real understanding of the underlying complex molecular mechanisms, which frequently include interaction with the surrounding healthy tissue, the immune system, chronic inflammation. Heterogeneity is a major concern in many solid tumors [4]. Studying spatial and temporal diversity of the cancer genome is chiefly realized by multiregion sequencing. Sequencing technology evolution includes traditional Sanger sequencing, single nucleotide polymorphism (SNP) arrays [5], genome-wide analysis of somatic copy-number alterations (SCNAs) [6], and massive parallel sequencing, frequently labeled as “next-generation” [7]. High-throughput sequencing can generate large amounts of data concerning the lung cancer exome and genome in order to discover new mutations, structural variations [8], microRNAs [9], and analyze DNA methylation [10]. Deep sequencing methods aim for high numbers of unique reads in every region of a sequence, thus raising the sequencing accuracy [11]. Discovering the molecular profile of a lung tumor is of utmost importance in the era of patient-tailored therapy. Markers such as epidermal growth factor receptor (*EGFR*), anaplastic lymphoma tyrosine kinase (*ALK*), and ROS proto-oncogene 1 receptor tyrosine kinase (*ROS1*) gene rearrangements not only guide the therapy, but also give important prognostic insights [12]. Future therapies might also target epigenetic changes and reprogram tumor cells by making them more vulnerable. This is called epigenetic priming. Since epigenetics play a crucial role in lung tumorigenesis and progression, affect multiple pathways, and regulate several cancer cell properties, targeting the epigenome could bring important changes in therapy [13,14].

Recent data report a branched evolution of non-small-cell lung cancer (NSCLC) with heterogeneous mutations that develop in response to local environmental factors [15]. This type of evolution is thought to be partially responsible for treatment resistance, as subclones with possible resistance mutations are not hindered from developing. The present biopsy techniques have several disadvantages, evident particularly in patients with acquired resistance to therapy, such as invasiveness, potential complications, problematic repeatability, and impossibility to supply data on all cancer subclones. Important studies in the field, such as IPASS (Iressa Pan-Asia Study) and INTEREST (Iressa NSCLC Trial Evaluating Response and Survival again Taxotere), report that only 42% and 31%, respectively, of lung cancer patients have enough biopsy tissue for a complete molecular diagnosis [16,17]; therefore, in many cases, multiple tissue biopsies are required for a full tumor characterization. 

Liquid biopsy is an emerging technique that might be able to bring important contributions to precision medicine by easily gathering robust and reproducible data. It implicates isolation of tumor-derived components in peripheral blood, saliva, urine, or other bodily fluids, and genomic or proteomic evaluation. The first study that mentioned the existence of circulating nucleic acids was published in 1948 [18] and opened a new chapter in understanding of disease progression. Since then, it has been proven that cell-free DNA (cfDNA, with its subset, circulating tumor DNA—ctDNA), mRNA, microRNA (miRNA), and long noncoding RNAs (lncRNAs) can provide useful information regarding tumor burden, treatment responsiveness, and malignant progression [19]. 

Therefore, detecting tumor markers in circulation as a minimally invasive diagnostic procedure in order to guide early diagnosis, to assess resistance detection and determine new driver mutations can shift the paradigm in the treatment of patients with lung cancer in an advanced stage. Currently, liquid biopsy is not incorporated in the clinical routine, but its potential is rapidly developing. Herein, we briefly present the complexity of lung tumorigenesis, with a particular emphasis on the implication of noncoding RNAs, and how liquid biopsy could change the management of lung cancer, together with the challenges that this method implies. 

## 2. Molecular Genetics of Lung Tumors 

Understanding lung carcinogenesis is an important step in discovering useful diagnostic biomarkers. The highly specialized structure of the lung contains secretory cells situated between the pseudostratified epithelium of the airways. These cells contribute to the repair process that follows epithelial injury. On the alveolar level, there are two types of cells: type I, the dominant one, involved in gas exchange, but unable to replicate, and type II, responsible for production of pulmonary surfactant, precursors of type I alveolar cell in the process of repair and having the capability to divide [20]. 

The original cells for cancers of the lung are known to be involved in airway repair processes. For Kirsten rat sarcoma 2 viral oncogene homolog (*KRAS*)-driven adenocarcinomas, the most common type of lung adenocarcinomas, the initiating cells were found to be the alveolar type II epithelial cells [21]. Squamous cell carcinoma most often origins from the bronchial epithelium of central, larger bronchi and has morphological features such as intercellular bridges, squamous pearl formation, and individual cell keratinization [22]. Using mouse models and expression levels of neuroendocrine markers, it is practically established that in small cell lung cancer (SCLC), the tumor process starts from neuroendocrine cells [23]. Table 1 shows common somatic mutations present in the major histological lung cancer types. 

Chronic inflammation and cancer are hypothesized to be interconnected, with inflammation being considered one of the most significant epigenetic factors in epithelial cancer development. Approximately 20% of tumors are considered to be related to persistent inflammation. Oncogenic mutations lead to activation of inflammatory pathways, creating a tumor microenvironment composed of chemokines and cytokines such as interleukins (IL)-1α, Il-6, Il-8, TNFα, or the CXCL12–CXCR4 axis. These factors are able to promote cell growth, angiogenesis, and invasion [35]. 

The role of epithelial NF-κB signaling pathway activation has been an object of study in many cancers, including esophageal, breast, prostate, and gastric tumors. This key transcription factor regulates many signaling pathways implicated in cancer promotion and was found to be activated by tobacco exposure in lung cancer cell lines. An interesting consequence of the occurrence of high levels of NF-κB expression in epithelial tumors is that NF-κB was found to inhibit chemotherapy-induced apoptosis and might therefore cause treatment resistance. Authors suggest a future basis for targeting NF-κB activation in therapeutic approaches of lung cancer [36]. Saxon et al. (2018) most recently demonstrated, using gene expression microarray analysis, that activation of the NF-kB subunit 2 (p52) is a mediator in the development of lung cancer, as it promotes proliferation, and could also be taken into consideration as a new therapeutic target in the future [37]. Recently, Wang et al. (2018) identified a new function of the macrophage Nitric Oxide Synthase 2 (NOS2), which consists of promoting squamous cell carcinogenesis in lung growth by promoting the migration and survival of macrophages, decreasing lipid metabolism and maintaining the inflammatory microenvironment [38].

The interactions presented in this section are not intended and are impossible to be exhaustive. They are illustrations of the fact that lung cancer development is under extensive research, with the purpose of finding new biomarkers useful in the prevention and management of lung cancer. 

## 3. Noncoding RNAs in Lung Cancer

Many years after the discovery of the genetic information at cellular level, scientists have been mainly focused on the housekeeping and the messenger role of RNA between genes and proteins. The knowledge that almost 98% of the human genome is not translated into proteins, but apparently most of it is transcribed into RNA, has raised curiosity regarding how these noncoding molecules could influence biological processes. Soon it has been identified that these transcripts, encoded by the so-called “junk” DNA or “dark matter”, are key regulators involved in several essential processes, including cell cycle control, regulation of gene expression, apoptosis, chromatin remodeling, and epigenetic modification [39]. 

The noncoding RNAs (ncRNAs) can be divided into short noncoding RNAs, consisting of a sequence of less than 200 bp, and long noncoding RNAs with more than 200 bp [40]. The group of short ncRNA comprises molecules such as microRNA (miRNA), piwi-interacting RNA (piRNA), small interfering RNA (siRNA), and small nucleolar RNA (snoRNA), which are characterized by their size, biogenesis, silencing mechanisms, and interactions with proteins from the Argonaute protein superfamily. The category of long ncRNA was recently found to include transcripts, such as transcribed ultraconserved region (T-UCR), natural antisense transcript (NAT), and telomerase RNA component (TERC). This last category of ncRNAs also includes molecules with coding potential, namely pseudogene transcripts and circular RNAs (circRNAs). In contrast to miRNAs, they have a crucial role in the control of gene expression during development and organogenesis, both in the cytoplasm and nucleus, and can serve as precursors for smaller ncRNAs [40,41].

The ncRNAs can influence lung tumorigenesis and proliferation and have therefore attracted attention as future possible biomarkers or therapeutic targets [42]. With the persistent low survival of patients with lung cancer and the fast onset of chemoresistance comes the necessity to find other druggable molecules. New technologies, such as RNA sequencing, along with bioinformatics, can facilitate the identification of new promising markers and bring more light in the field of molecular tumor biology. 

## 4. MicroRNAs

MicroRNAs are small noncoding RNAs with a size ranging from 19 to 25 nucleotides that have received most of the attention from the scientific community for being involved, among other mechanisms, in initiation, progression, and dissemination of cancer. These processes happen mainly by post-transcriptional alteration of gene expression. miRNAs can act both as oncogenes and tumor suppressor genes and their different expression level in normal and tumor tissue has been documented, including in lung cancer [43]. Their biogenesis is a multistep process that starts with pri-miRNAs, precursors with a length of more than 100 nucleotides. pri-miRNAs can be transcribed from its own promotor from intronic, intergenic, or polycistronic loci or as part of a single hairpin. The precursors are first transcribed by RNA polymerase II and then intranuclearly processed by Drosha, an RNase III enzyme, and Pasha, a double-stranded RNA (dsRNA)-binding protein to form the pre-miRNA. This product is transported to the cytoplasm, where it is cleaved by Dicer, another RNase III enzyme. The endoribonuclease removes the loop that joins the 3′ and 5′ arm and forms the miRNA: miRNA duplex. The duplex unwinds and one strand assembles into RNA-induced silencing complex (RISC), where it plays the role of a template that identifies the complementary mRNA and downregulates its expression by either direct degradation or translational repression. The other strand of the duplex is degraded. 

Accurate miRNA expression is mainly regulated on the transcriptional level. Regulation of Drosha, Dicer, and RISC can also affect global miRNA expression [39]. miRNAs are involved in numerous genomic and epigenomic processes, where one single miRNA can have numerous targets, whereas one target can be regulated by several miRNAs. 

In 2011, Hanahan et al. published a review where they tried to simplify the substantial complexity of cancer biology by establishing eight hallmarks of cancer [44]. Research in lung cancer has reported miRNAs involved in seven out of eight hallmarks. Table 2 summarizes the main miRNAs that play a role in each of the eight hallmarks.

The most important and known signaling pathway involved in sustained cell proliferation is the EGFR-signaling pathway, which together with PI3K/Akt/mTOR and Ras/Raf/MEK/ERK pathways enhances cell growth and proliferation [45]. *EGFR* is a direct target for a lot of miRNAs, such as miR-7, miR-34, miR-128, miR-145, miR-146, miR-128, miR-542-5p [45]. Studies on NSCLC cell lines found that miR-760 can reduce cellular proliferation by suppressing *ROS1* expression [47]. *KRAS*, which is another activator of the Ras/Raf/MEK/ERK pathway, is repressed by the let-7 family, slowing the in vitro growth of lung cancer cells [48].

Evading growth suppressors is mainly caused by deregulation of pathways involving p53 and retinoblastoma (RB) proteins. Two of the miRNAs involved in cell cycle arrest and cell senescence in lung cancer are miR-641 and miR-660 that target MDM2, a suppressor of p53 pathway [55,58]. miRNAs, such as the miR-200 family, have been discovered to play a role in the complex invasion–metastasis process in lung cancer, at the center of which lies the epithelial-to-mesenchymal transition (EMT), characterized by increased cell motility caused by the loss of cell adhesion mediated by E-cadherin [59]. Another hallmark of cancer is angiogenesis induction, with well-known participants, such as vascular endothelial growth factors (VEGFs) and the Akt pathway. Family members of miR-200, miR-126, and miR-128 can directly target VEGFs and inhibit angiogenesis in lung cancer cell lines [60,61,62], while miR-494 activates the Akt pathway by targeting the phosphatase and tensin homolog (PTEN) [63]. 

In order to sustain unrestrained proliferation, tumor cells can deregulate their metabolic pathways. miR-144 is downregulated in lung cancer cells, generating an upregulated expression of the glucose transporter (GLUT1) with amplified glucose uptake [65]. miR-31-5p can increase lung cancer spreading by downregulating the inhibitor of hypoxia-inducible factor-1α (HIF-1α) and enhancing glycolysis [76]. Furthermore, miRNAs are involved in the interaction between the immune system and cancer cells by regulating the PD-L1 expression and evading immune destruction. miR-197 expression levels are inversely associated with PD-L1 levels and are linked to worse overall survival in NSCLC patients [71]. Avoiding programmed cell death, or apoptosis, is another hallmark of cancer and occurs by caspase activation leading to cell destruction. Wang et al. reported that overexpression of miR-16-1 in the human NSCLC A549 cell line can inhibit cell proliferation and apoptosis by controlling the expression of BCL2, CDKN1B, BAX, and CASP3 [72].

## 5. Long Noncoding RNAs

Long noncoding RNAs (lncRNAs) are functional RNA molecules that have emerged in the field of tumor biology as a novel class of possible biomarkers and druggable targets. They constitute a diverse group of RNA molecules with a length of more than 200 nucleotides each. There is accumulating evidence that lncRNAs can control key pathways involved in lung tumorigenesis, including cell proliferation and metastasis. What makes them attractive as therapeutic targets is the fact that they are frequently expressed in a disease- or developmental stage-specific way. However, knowledge about their specific roles in cancer is still incomplete [77]. This chapter tries to provide an overview the lncRNAs known to be involved in the development of lung cancer.

Among the most studied lncRNAs are HOTAIR (HOX transcript antisense RNA) and H19. These molecules play an important role in transcriptional and posttranscriptional processes by reducing target gene expression. [78]. The role of lncRNA HOTAIR in epigenetic gene regulation has led the scientific community to evaluate its function in cancer. A high expression of HOTAIR was associated with an advanced stage of NSCLC, metastases, poor survival of patients, and a shorter disease-free interval after surgery. Downregulation of HOTAIR mediated by siRNAs decreases in vitro migration and invasion of NSCLC cells [69]. Deregulation of H19 expression at a genomic, epigenetic, transcriptional, or posttranscriptional level might enable its oncogenic potential. In support of its oncogenic role, Chen et al. found an association between overexpression of MDIG (mineral dust-induced gene) and H19 and poor survival in smokers with lung cancer [79]. Another study considered the relationship between H19, TP53, and HIF-1α and found that only a combination of absent or nonfunctional TP53 and high HIF-1α levels can increase expression of H19, pointing toward an oncogenic character of H19 in the context of cancers that harbor a TP53 aberration [80].

MALAT1 (metastasis-associated lung adenocarcinoma transcript 1) is one of the first cancer-related lncRNAs that was originally recognized as a marker of survival and metastasis in patients with early stage NSCLC [81]. Its expression is a negative prognostic factor in patients with SCC [82] and elevated levels have been associated with lung cancer brain metastases. Studies on mice have found that depletion of MALAT1 in normal cells could be tolerated, which can make MALAT1 a potential drug target for the prevention of metastasis in lung cancer [68]

In summary, lncRNAs might play a part in most of the biological processes involved in lung cancer tumorigenesis and disease progress. Roth et al. extensively reviewed the lncRNAs involved in all aspects of lung cancer genesis and progression. For example, lncRNAs such as MALAT1, HOTAIR, MEG3, GAS5, SPRY4-IT, ANRIL, and BANCR might be involved in resisting cell death. Others support cell metastasis by influencing invasion (MALAT1, HOTAIR, SPRY4-IT, CARLO-5, CCAT2, PVT1, BANCR, MVIH), migration (MALAT1, HOTAIR, ANRIL, SPRY4-IT, CCAT2, PVT1, BANCR, MVIH, GHSROS), and epithelial–mesenchymal transition (MALAT1, SPRY4-IT, CARLO-5, BANCR) [82]. The clinical integration of lncRNA knowledge in terms of predictive biomarker signatures and possible therapeutic targets could benefit patient survival and prognosis. Figure 1 portrays the main lncRNAs involved in the diagnosis and prognosis of NSCLC. 

One of the latest directions in the study of the mechanisms behind lung cancer development and progression has unraveled the complex interactions between miRNAs and lncRNAs. It seems that a type of lncRNAs, competitive endogenous RNAs (ceRNAs), can regulate microRNAs, by competing for binding to miRNAs with mRNAs and inhibiting miRNA expression. miRNA–lncRNA interactions have been found to influence epithelial–mesenchymal transition (EMT) and the classic tumor-associated pathways, contributing to tumorigenesis, metastatic spread, and chemoresistance in lung cancer [77]. Since a lot of classic signal pathways that are regulated by miRNAs, such as JNK [83], Wnt/β-catenin signaling pathway [84], or MAPK [85], also influence the expression of several lncRNAs in lung cancer tissues, it is safe to presume that some of the classic tumor-associated pathways are coregulated by lncRNAs and miRNAs. For example, miR-21 influences GAS5′s regulation of NSCLC cisplatin sensitivity through the PTEN pathway [86], and lncRNA-XIST can promote proliferation and metastases in lung cancer via miR-140 regulation of the p53 (inhibitor of apoptosis-stimulating protein of p53—iASPP) pathway [87]. 

## 6. Liquid Biopsy in Lung Cancer

Detecting circulating tumor-derived nucleic acids and proteins in patients with advanced stage lung cancer has started to gain clinical utility in recent years. ncRNAs have been acknowledged to be closely linked to tumorigenesis, proliferation, and metastasis; therefore, circulating miRNAs and lncRNAs could serve as biomarkers in lung cancer diagnosis, therapy, and prognosis. Properties such as stability, detectability, and resistance to RNAse degradation might turn these molecules into reliable cancer biomarkers. 

New target therapies, such as *EGFR*-activating tyrosine kinase inhibitors (TKI), confirmed their superiority to chemotherapy and research focused especially on *EGFR*-activating mutations and abnormal gene fusion between echinoderm microtubule-associated protein-like 4 and anaplastic lymphoma kinase (*EML4-ALK*) [24,25,87,88,89]. In order to detect the presence of mutations, repeated invasive examinations are required, which can cause discomfort to the patient. These medical procedures, including computed tomography (CT)-guided biopsy and bronchoscopic or surgical biopsy of the lung, can present certain impediments that could be overcome by liquid biopsy. Lack of sufficient tissue for molecular diagnosis [90], tumor heterogeneity [91], different failure rates of tumor genotyping methods used in routine clinical settings for the detection of *EGFR, ALK*, and *KRAS* [30], and associated risks and costs are some of the limitations that make liquid biopsy an appealing future surrogate of tissue biopsy in lung cancer management. 

At present, definite clinical validity and utility is yet to be established as only a few prospective trials have been carried out [92,93,94]. There is also limited data on how different patient-related factors could influence detection of circulating nucleic acids and how the resulting data, even though capable of representing heterogeneity, might be influenced by differential tumor cell turnover [95]. The methods and assays that allow detection and analysis of tumor-derived nucleic acids can be divided into two categories—the ones targeting a single or limited number of mutations, such as real-time or digital polymerase chain reaction (RT-PCR, dPCR), and those that can detect a large number of genes, such as microarray technologies or next-generation sequencing (NGS) [96]. In lung cancer, targeted gene assays are used for detecting *EGFR* variants in NSCLC with a reported concordance between tissue and plasma of 70% to 90% [97,98,99]. It is still uncertain whether biological or analytical aspects cause discordance between detection of a somatic variant in the tissue biopsy, but not in ctDNA assays. Mutations in cfDNA can also be found in healthy subjects, particularly with increasing age. Moreover, in a recent study, researchers have reported that cfDNA levels have a circadian variation both in healthy subjects and in lung cancer patients, with significant decline during the day [100]. In order to overcome these disadvantages, future studies need to evaluate reference materials with known mutations that would allow evaluation of analytical performance of the investigation independently of other factors that could influence the comparison between tissue and blood specimens. Usage of such standardized samples up to now has made it possible to determine the lower limits of different assays involved in detection of single mutations. Optimal limits are yet to be established [95]. 

Current evidence supports the use of plasma for optimal detection of ctDNA. This is mainly determined by the fact that serum lacks clotting factors and has increased amounts of normal DNA, which causes a dilution of ctDNA, resulting especially from leukocyte destruction that occurs during clotting [101]. ctDNA represents <1% of total cfDNA, with sizes between 180 bp and 1000 bp (originated from apoptosis) or 10,000 bp (originated from tumor necrosis, see Figure 2) [102], making PCR-based assays the best method for detection and genotyping. 

Cells exchange different vesicles such as oncosomes, microvesicles, exosomes, ectosomes, and lipoproteins. Extracellular vesicles (EV) are another origin of nucleic acids, containing fragments of >10 kb of DNA that carry EGFR, p53, or KRAS mutations [103]. The large size of EVs makes them an easier target to detect, with a number of recent studies that support their use as complementary biomarkers in lung tumor assessment [104,105]. These vesicles reflect the cell-to-cell communication, and the active process of secretion from tumor cells can create a blood concentration of up to >10^9^ vesicles/mL, which is a lot more compared to circulating tumor cells (CTCs) that vary between 1 and 10 CTCs/mL of blood. For example, oncosomes are large plasma membrane-derived EVs spontaneously released by tumor cells that present metalloproteinases with invasive properties [106]. Castellanos-Rizaldos et al. (2018) developed and validated a test that detects the EGFR T790M mutation on exosomal RNA/DNA and cfDNA, obtaining a sensitivity of 92% and a specificity of 89%, with a high sensitivity for patients without extrathoracic metastases, for whom liquid biopsy presented challenges [107]. 

Liquid biopsy may find use mainly in the management of patients with advanced stages of lung cancer, in treatment selection for these patients, and noninvasive monitoring, but also for detection of residual disease in early stages. Figure 3 tries to show the origin of the circulating tumor-derived nucleic acids in the liquid biopsy. These biomarkers have the potential to reduce mortality in lung cancer, especially when considering that there is no harm by radiation, as in the case of low-dose computed tomography (LDCT). Use of ctDNA for screening and treatment decision in late stages of lung cancer still needs prospective clinical trials to test its utility as a stand-alone diagnostic test, and studies to demonstrate that ctDNA assays can provide the same data as genomic evaluation of tissue samples. Early results from the bioMILD trial suggest that using LDCT accompanied by a blood microRNA assay is effective and safe in screening for lung cancer.

Monitoring treatment response by repeated quantitative analysis of ctDNA is very appealing in terms of patient comfort and health-associated costs. Tumor-associated proteins, such as cytokeratin 19 fragment (CYFRA 21-1) and carcinoembryonic antigen (CEA) for NSCLC [108] or neuron specific enolase (NSE) for SCLC [109], have proved their benefit only in addition to clinical, radiological, or histological suspicion of lung cancer and only on a qualitative level. Evaluating tumor burden, even though attractive, raises technical challenges in regard to efficiency, reproducibility, and validation of different assays, also requiring uniformity between laboratories, so that trials can be comparable. Researchers have tried to correlate prognosis and treatment responses of lung tumors with ctDNA levels; however, the studies are small and mostly retrospective, with no certain proofs of better patient outcomes [110,111,112,113]. For patients with early-stage cancer, liquid biopsy could ease detection and monitoring of residual disease after surgical treatment. Research in this particular area has been limited, since ctDNA generally has a lower rate of detection than in patients with advanced disease [114]. Abbosh et al. (2017) [115] demonstrated that persistent ctDNA after curative therapy of lung cancer predicts a high risk of recurrence. Unfortunately, none of the studies have used imaging in addition to plasma to establish if overt metastatic disease has not developed at the moment of blood testing.

## 7. Circulating Noncoding RNAs 

Until recently, the majority of the research has examined ncRNAs from patient tissues obtained by invasive methods. Liquid biopsy could have the potential to change the landscape of lung cancer research, and subsequently its diagnostic and management, by detecting circulating molecules such as miRNAs and lncRNAs.

miRNAs can regulate protein translation and are released by both normal and tumor cells in plasma [116], among other biological fluids. Circulating miRNAs can reproduce the interactions that take place in the tumor microenvironment [117], and as they are more resistant to degradation, they could prove to be useful biomarkers in lung cancer detection. Reverse transcription quantitative PCR (RT-qPCR), microarray platforms and NGS can be employed in the detection of circulating miRNA. With NGS being still quite expensive and microarray a nonquantitative method with necessary experimental validation, RT-qPCR followed by TaqMan PCR is currently the most widely used assay for detection of circulating miRNA. The disadvantage of this technology is that, despite being accurate and sensitive, it can only detect already known miRNAs [118]. 

In a systematic review of 20 studies, Moretti et al. (2017) sought to identify circulating miRNA suitable for stage I-II NSCLC screening and found miR-223, miR-20a, miR-448, and miR-145 to have >80% sensitivity and the area under the receiver operating characteristic curve (AUROC) to be >0.80. The authors also detected >90% specificity for miR-628-3p, miR-29c, miR-210, and miR-1244. Using the above panel of miRNAs with high sensitivity, followed by the second panel with high specificity in screening for stage I-II NSCLC, they obtained a pooled specificity of 93.4% and a pooled sensitivity of 91.6% [119]. However, these results, among those from other studies [120,121], need proper validation before they can prove their benefit. In a recent study, miR-223 has been validated as an effective and reproducible serum biomarker of early stage NSCLC by using digital-droplet PCR technology [122]. Li et al. (2017) identified in a prospective study, using RT-qPCR on plasma samples from 71 patients (41 *ALK*-positive), a panel of three circulating miRNAs that can discriminate between patients diagnosed with *ALK*-positive and *ALK*-negative lung cancers. It was also shown that miR-660-5p and miR-362-5p could possibly be used in predicting response to crizotinib, but again, these results require high-power validation studies and comparison of miRNAs in plasma and tissue from the same patient [123].

lncRNas have also been identified in the circulatory system of lung cancer patients. Three studies reported expression of lncRNAs in whole blood samples, measured by RT-PCR or qRT-PCR. Two of the studies found that the expression of MALAT1 in whole blood samples was lower in patients diagnosed with NSCLC than in the rest of the population, in contrast to the upregulated expression of MALAT1 in the tissues of NSCLC patients [124,125]. Another study found elevated SCAL1 expression in the airways of smokers and several lung cancer cell lines [126]. As for the detection of lncRNAs in the serum of NSCLC patients, one study reported elevated levels of XIST and HIF1A-AS1 detected in both the tumors and serum of these patients, with a significant decrease in expression after surgical resection of the tumors [127]. SOX2OT and ANRIL were found to be reliable diagnostic and prognostic biomarkers for NSCLC, by using combined tissue and serum detection [128]. Significant increases in the plasma expression of lncRNA-UCA1 [129], lncRNA16 [130], SPRY4-IT1, ANRIL, and NEAT1 [129] indicate the biomarker potential that these lncRNAs might prove to have in NSCLC patients. One recent study showed a significant upregulation of SOX2OT expression level when matched to controls, and a correlation with the TNM stage and histology. Moreover, the authors reported an important downregulation of plasma GAS5 expression levels in NSCLC patients correlated with the TNM stage and tumor differentiation. This comes in support of another study that reported decreased GAS5 expression levels in the plasma of 90 NSCLC patients [131]. 

Even though circulating ncRNAs are easily accessible, which makes them perfect for use as biomarkers, reproducibility among studies is reported to be low, and specificity to a particular pathology is difficult to attain, since there is currently scarce information about the influence that various diseases and environmental factors have on the levels of ncRNAs.

## 8. Detecting EGFR, ALK, and ROS1 in Plasma

At present, clinical validity for PCR-based ctDNA testing has been recognized by the FDA (Food and Drug Administration) in the United States and by the EMA (European Medicines Agency) in Europe, and has been approved for *EGFR* mutation detection in patients with advanced NSCLC. Research has demonstrated that although positive *EGFR* results can be reliably used in treatment decisions, a negative result should be confirmed by tissue biopsy [132]. Mao et al. (2017) reported in a meta-analysis of 15 studies a pooled sensitivity of peripheral blood in detecting *EGFR* mutation status in advanced NSCLC patients of 0.69 (95% CI: 0.59–0.78) and a pooled specificity of 0.97 (95% CI: 0.94–0.99), with a summary receiver operating characteristic curve of 0.93 (95% CI: 0.91–0.95) [99].

Four meta-analyses, of which one was mentioned earlier [99], sought to determine whether ctDNA can have the same diagnostic value as tissue biopsy. Qian et al. (2016) reported the highest specificity using the amplification-refractory mutation system (ARMS) [133] and came in support of another meta-analysis of 27 studies conducted by Qiu et al. (2015) which concluded that ARMS had the highest clinical utility and diagnostic accuracy [134]. In terms of sensitivity, Luo et al. (2014) detected denaturing high-performance liquid chromatography (DHPLC) and resolution melting analysis (HRM) as being better than ARMS [135]. These studies, along with two multicenter diagnostic trials [136,137], demonstrated a high specificity of circulating EGFR-mutated DNA and a low sensitivity between 40% and 70%. 

Large retrospective studies tried to evaluate the role of circulating *EGFR*-mutated DNA in detecting TKI-resistant genetic aberrations. Although there is a significant rate of response to *EGFR*-TKIs, a large percentage of patients develop resistance to treatment, with a progression free survival (PFS) between 9 months and 14 months [138]. Once the T790M mutation has been established as the most common determinant of resistance and selective drugs such as osimertinib have proved their efficiency for patients harboring this genetic alteration, the need for a noninvasive way of testing for T790M has increased significantly [139]. A number of studies reported applicability in detecting T790M in blood samples and an ability to correlate this finding with treatment response [140,141,142]. Oxnard et al. (2016) conducted a study on 216 patients and found that plasma genotyping for T790M using beads, emulsions, amplification, and magnetics (BEAM) had a 30% false-negative rate [140].

Research involving lung cancer genotyping for other mutations, such as rearrangements in *ALK*, *ROS1*, or *RET* genes, amplifications in the *MET* gene, or mutations in the *BRAF* or *HER2* genes, has started to slowly develop. Rolfo et al. (2017), in a small study on 18 patients, analyzed the possibility of detecting *EML4-ALK* translocation in exosomes using NGS and found a 100% specificity and a 69% sensitivity of the method [143]. Studies involving detection of PD-L1 status [144] and MET amplification [145] by liquid biopsy have also shown promising results.

A potential problem of detecting mutations in plasma could be that the patient population included in the research may not be representative of those for whom this method would provide clinical use. Since the main need for liquid biopsy comes from the impossibility to obtain enough tissue for a molecular diagnosis, the fact that these studies did not include patients with this characteristic is a major disadvantage. Only one prospective trial has evaluated the prognosis of lung cancer patients when the target therapy was solely designed based on a liquid biopsy result [146]. 

## 9. Conclusions

Tissue continues to “be the issue” in lung cancer, but current advancement in molecular analysis can find new ways to better diagnose and treat oncological patients. Liquid biopsy might be an important and promising approach in the management of patients with advanced lung cancer. Advantages, such as a less invasive approach, repeatability, a wide insight into the tumor heterogeneity and genomic alterations, and the possibility of easily monitoring drug response, have led scientists to start thorough research with great progress in the field of liquid biopsy in the last five years. Tracking tumor changes and resistance in real time by using blood assays could avoid delays in therapy and increase patient survival. 

Circulating tumor-derived nucleic acids, such as ctDNA, mRNA, and ncRNAs, can offer insight regarding tumor burden, disease progression, metastasis, and therapy responsiveness. miRNAs and lncRNAs have recently emerged as functional RNA molecules that have changed the current understanding of tumorigenesis and might play an essential part as biomarkers and therapy targets. Identification of these molecules in the circulatory system using high-throughput sequencing methods has been successfully reported, especially when using combined panels of several molecules. The value of ncRNAs in the liquid biopsy comes from the fact that these promising entities are stable to RNase degradation and detectable not only in blood, but also in other bodily fluids. 

Detecting tumor markers such as *EGFR, ALK, ROS1*, and PDL-1 in the blood or plasma might not only guide management, but also detect early therapy resistance, relapse, and establish a prognosis.

The “holy grail” in the field of liquid biopsy still remains the development of a test that could detect tumor-derived nucleic acids before clinical symptoms arise. Using this method as screening could allow the much-wanted early diagnosis for lung cancer patients. However, the need to perform this type of blood test in apparently healthy individuals still requires assessment in terms of the impact that it could have on the natural history of tumors and its practicality. 

Liquid biopsies are seldom implemented nowadays in routine testing, despite remarkable advances made in the past years. A lot of effort has been made to overcome technical difficulties and find standardization among the detection procedures. Unfortunately, despite the current enthusiasm, with small exceptions, there is a lack of clinical validity and utility supported by robust research. Tumor genotyping is a rapidly developing research area and there is a high probability that new evidence will emerge that will help in reaching a consensus about, among other factors, the lack of study reproducibility and the variation among different detection techniques. This will eventually unleash the true potential of liquid biopsies. 

## Figures and Tables

**Figure 1 diagnostics-09-00216-f001:**
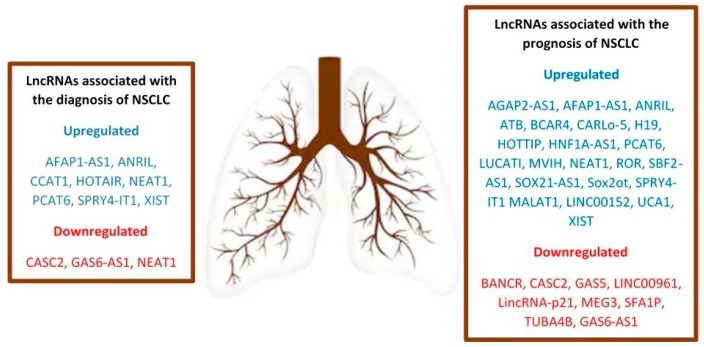
Main long noncoding RNAs (lncRNAs) related to the diagnosis and prognosis of non-small-cell lung cancer (NSCLC).

**Figure 2 diagnostics-09-00216-f002:**
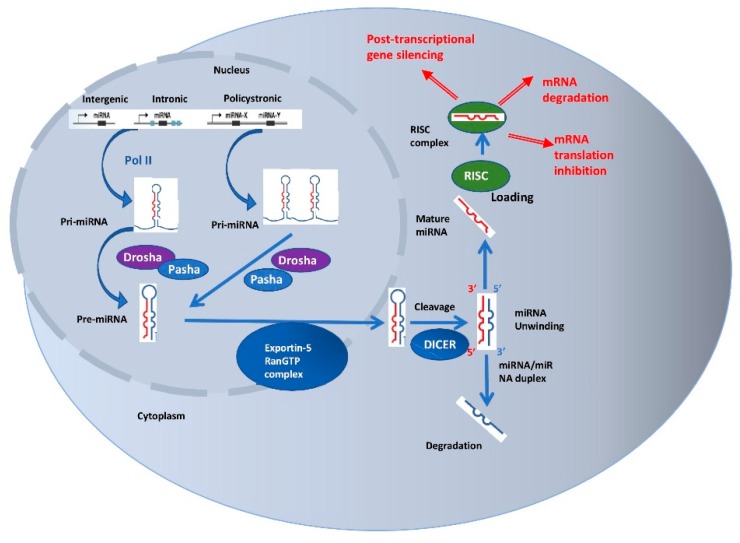
Canonical miRNA genesis and processing pathway. The miRNA gene is transcribed in the nucleus from intronic, intergenic, or polycistronic loci by RNA polymerase II or III and forms a transcript that is called primary miRNA (pri-miRNA). pri-miRNA hairpins are double-stranded RNA (dsRNA) structures cleaved by Drosha, an RNAse III-type enzyme, and Pasha, also known as DiGeorge syndrome critical region gene 8 (DGCR8), to form a 70–100 nucleotide long precursor (pre-miRNA). Pre-miRNA hairpin is transported to cytoplasm by the exportin-5 and RanGTP cofactor and then processed by the Dicer complex, another RNase III enzyme, into a miRNA:miRNA duplex. The unwinding of the duplex forms the 18–23 nt mature miRNA. One strand of the duplex binds to Ago and forms RNA-induced silencing complex (RISC). Once loaded, the RISC can target and induce mRNA negative expression by cleavage, translational inhibition or destabilization and degradation.

**Figure 3 diagnostics-09-00216-f003:**
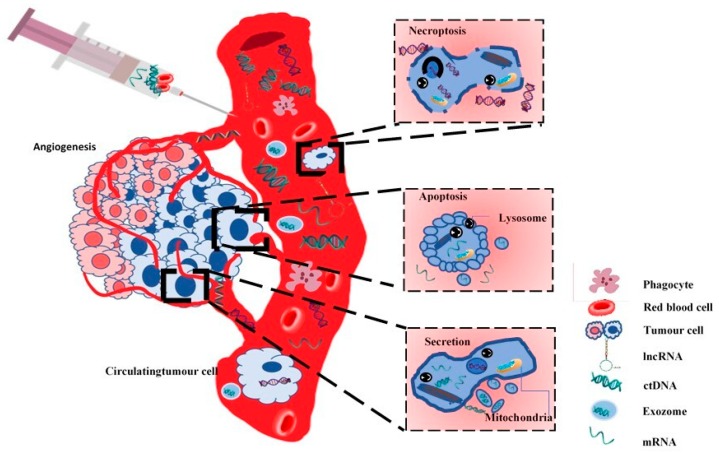
The origin of circulating tumor-derived cell-free nucleic acids—mutant DNA and RNA, which includes mRNA and noncoding RNA. cfDNA originates from different forms and levels of apoptosis or necrosis of healthy or diseased tissue or from extracellular vesicles (EVs) secreted by tumor cells. The genetic alterations detected in blood include point mutations, structural rearrangements, copy number alterations, and microsatellite alterations, and methylation of DNA/RNA.

**Table 1 diagnostics-09-00216-t001:** Examples of genetic abnormalities in the main lung tumor types.

Histological Type of Lung Cancer	Tumor Suppressor Genes	Oncogenes	References
Adenocarcinoma	*TP53* Mt 90% 17p LOH 70% 9p LOH 50–75% 3p allele loss 50–80% 13q LOH 40–60% *CDKN2A* 10–40%	*KRAS* Mt 32.2% *EGFR* Mt 15% *FGFR4* Mt 7% *EML4/ALK* Fus 1–5% *BRAF* Mt 2–3% *MET Mt/Amp* 1–20% *Her2* Mt 0.9%	[12,24,25,26,27,28]
Squamous cell carcinoma		*FGFR1* Amp 20% *PIK3CA* Mt 12% *KRAS* Mt 6% *EGFR* Mt 5% *BRAF* Mt 2%	[18,24,25,26,29,30,31]
SCLC	17p LOH 80–90% 13q LOH 75% 9p LOH 20–50% 3p allele loss >90% *TP53* Mt 90% Absent *RB1* expression 90%	*KRAS* Mt 25% *RET* Fus 5% *EGFR* Mt 4% *ROS1* Fus 1% *PIK3CA* Mt 3% *MYC* Amp 15–30%	[25,32,33,34]

Amp = amplification; Fus = fusion; Mt = mutation; LOH = loss of heterozygosity.

**Table 2 diagnostics-09-00216-t002:** Most important miRNAs involved in hallmark capabilities of lung cancer.

Hallmark of Cancer	Target of miRNA	miRNA	Reference
Sustaining proliferative signaling	*EGFR* *ROS1* *ALK* *KRAS* *PI3K*	miR-7, miR-27a-3p, miR-30, miR-34, miR-128, miR-133, miR-134, miR-145, miR-146, miR-149, miR-218, miR-542-5p, miR-760, miR-96, let-7, miR-193a-3p, miR-181a-5p, miR-148a-3p (via SOS2), miR-1258 (via GRB2), miR-520a-3p	[45,46,47,48,49,50,51,52,53,54]
Evading growth suppression	*MDM2/p53* *RB* *E2F*	miR-641, miR-660, miR-15/miR-16, miR-449a	[55,56,57,58]
Enabling replicative immortality		No miRNAs reported to target hTERT in lung cancer studies	
Invasion and metastasis	*ZEB1 and ZEB2*	miR-200	[59]
Inducing angiogenesis	*VEGF* *PTEN*	mir-200, miR-126, miR128, miR-494, miR-497	[60,61,62,63,64]
Deregulating cellular energetics	*GLUT1* *HIF-1α* *LDHA*	miR-144, miR-199a, miR-31-5p, miR-33b	[65,66,67,68,69]
Avoiding immune destruction and tumor-promoting inflammation	*PD-L1*	miR-34a, miR-197, miR-200 family	[59,70,71]
Resisting cell death	*Bcl-2*	miR-16-1, miR-130b (via PPAR-γ/VEGF)	[72,73]
Genome instability and mutation	*SPTAN1* *P53/PIK3R1 and CDC42*	miR-128-3p, miR-125b, miR-504, miR-29	[74,75]

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
