# Peer review of "Noncoding RNAs and Liquid Biopsy in Lung Cancer: A Literature Review"

_diagnostics, 2019, doi:10.3390/diagnostics9040216_

Round 1

Reviewer 1 Report

General:

Harangus et al. present a review of the use of liquid biopsy in the diagnostics and treatment of lung cancer with emphasis on circulating non-coding RNA (microRNA, lncRNA etc.) and circulating (mutated) DNA. The manuscript is generally well written but might benefit from some improvements.

The NLST study has shown that early detection of lung cancer (by yearly screening with LD-HRCT) leads to a reduction in mortality (N Engl J Med 2011; 365: 395-409). The same should be true for the use of circulating biomarkers, especially when considering that there is no harm by radiation. I suggest to include a few more sentences regarding early detection by liquid biopsy (actually, in the conclusion – lines 472ff – you already point that out) and why there is, so far, no biomarker screening in clinical practice (lack of biomarker validation in prospective cohort studies with serial sampling that would facilitate the collection of pre-diagnostic samples of body fluids etc.; see, for example: Pesch et al., Biochimica et Biophysica Acta 2014; 1844: 874–883).

Details:

Line 31: … frequently …

Line 33: … curative therapies in an advanced … (no comma)

Lines 40-41: Another reason might be: A lack of the real understanding of the underlying complex molecular mechanisms, which frequently include interaction with the surrounding healthy tissue, the immune system, chronic inflammation etc.

Line 52: … could also be target epigenetic … (please delete “be”)

Line 54: Since epigenetics plays …

Line 79: … with a particular accent on … (do you mean “emphasis on …” ?)

Line 94: There are some newer papers on possible original cells for cancer subtypes. Please check the literature, e.g., Sanchez Danes, Nat Rev Cancer 2018;18(9):549-561; Yeh, Oncotarget 2019;10(38):3760-3806; Montoro, Nature 2018;560:319–324; Ferone, Cancer Cell 2016;30(4):519-532 etc.

Apparently, different cell types in the lung can give rise to SqCC.

Line 114: perhaps you could explain that p52 is a subunit of NF-kB: … that activation of the NF-kB subunit 2 (p52) is a mediator …

Line 123: Do you mean “miRNAs represent such (markers in) tissue, …” ?

Line 139: … long non-coding RNAs with more than 200 bp (no comma?)

Line 145: Do you mean “The category of ncRNAs …” ?

Line 169: Do you mean “… where it plays the role of a template that identifies …”

Line 221/Table 2: “… Bcl-2, p27, Bax and caspase 3” - please use standard names for the human genes: BCL2, CDKN1B (there are many genes/proteins with the name p27), BAX, CASP3

Lines 238 & 247: you use two different definitions of HOTAIR, one is sufficient (HOX transcript antisense RNA)

Line 282: Please explain the acronym iASPP (inhibitor of apoptosis-stimulating protein of p53). The official gene name of IASPP is now PPP1R13L (protein phosphatase 1 regulatory subunit 13 like)

Line 293: … EGFR-activating …

Line 299: What do you mean with “deficient genotyping techniques”? Nowadays, single-cell sequencing is possible (but, of course, not available in every lab).

Lines 310-312: You could also mention that even in healthy people (particularly with increasing age) one can find mutations in (circulating) DNA. The biggest problem, however, is the dilution of tumor DNA by that of normal tissue, as you point out in the following paragraph.

Another interesting factor that can lead to variations in biomarker levels in blood is circadian variation (Madsen et al., EBioMedicine 2019 Oct 21. pii: S2352-3964(19)30672-3).

Line 320: Do you mean “… that serum lacks clotting factors and (has increased amounts of) normal DNA, which causes a dilution …”?

Lines 322-323: You might add the size fraction 90 - 150 bp, which seem to be relevant (see Mouliere et al., Sci Transl Med 2018;10, eaat4921).

Line 324: Extracellular vesicles (EV) …

Line 331: … that detects the EGFR T790M mutation …

Line 339-340: I would suggest “… , copy number and microsatellite alterations, and aberrant methylation of DNA/RNA.”

Line 350: … (95% CI: 0.59-0.78) … (95% CI: 0.94-0.99) …

Lines 381ff: There is a recent update of the Moretti paper from the same group, where they confirmed miR-223: D'Antona et al., Cancer Epidemiol Biomarkers Prev 2019; 28(11):1926-1933.

Line 417: “… has been recognized by the ??? of the United States and Europe …”: perhaps you could insert the actual institution or authority (FDA etc.?) that did the recognition

Line 433: Once the T790M …

Line 469: the period at the end of the sentence is missing

Please double-check grammar and style.

Conclusion:

After addressing the above points, the manuscript by Harangus et al. can be recommended for publication in Diagnostics.

Author Response

We are very grateful for all the suggestions and for carefully and thoroughly reading our paper. Hopefully, we have addressed all the comments and suggestions and have modified the text accordingly.

We added comments on the side of the manuscript where changes were suggested.

Point 1. The NLST study has shown that early detection of lung cancer (by yearly screening with LD-HRCT) leads to a reduction in mortality (N Engl J Med 2011; 365: 395-409). The same should be true for the use of circulating biomarkers, especially when considering that there is no harm by radiation. I suggest to include a few more sentences regarding early detection by liquid biopsy (actually, in the conclusion – lines 472ff – you already point that out) and why there is, so far, no biomarker screening in clinical practice (lack of biomarker validation in prospective cohort studies with serial sampling that would facilitate the collection of pre-diagnostic samples of body fluids etc.; see, for example: Pesch et al., Biochimica et Biophysica Acta 2014; 1844: 874–883).

Response 1

We added some sentenced according to your suggestions. BioMild trial is one of the first trial that tries to introduce biomarker screening for lung cancer in clinical practice. The results were shared at the International Association for the Study of Lung Cancer (IASLC) 2019 World Conference on Lung Cancer (WCLC) (Abstract PL02.04).

Lines 348 – 256: “Liquid biopsy may find use mainly in the management of patients with advanced stages of lung cancer, in treatment selection for these patients and non-invasive monitoring, but also for detection of residual disease in early stages. Circulating biomarkers have the potential to reduce mortality in lung cancer, especially when considering that there is no harm by radiation, as it is in the case of low-dose computed tomography (LDCT). Use of ctDNA for screening and treatment decision in late stages of lung cancer still needs prospective clinical trials to test its utility as a stand-alone diagnostic test and studies to demonstrate that ctDNA assays can provide the same data as genomic evaluation of tissue samples. Early results from the bioMILD trial suggest that using LDCT accompanied by a blood microRNA assay is effective and safe in screening for lung cancer…

Point 2. Details

Response 2: We added comments in the text for each of your suggestions and modified the text accordingly.

Line 299: What do you mean with “deficient genotyping techniques”? Nowadays, single-cell sequencing is possible (but, of course, not available in every lab).

The article that we cited (Paul A. VanderLaan et a. ) evaluates the success and failure rate of EGFR mutation, KRAS mutation, and ALK FISH in a cohort of lung cancers subjected to routine clinical tumor genotype. It is true that single-cell sequencing is now possible, but we wanted to point out that sometimes the technique and type of tissue acquisition used routinely can lead to failure of mutation detection.

We modified the text according to the other minor comments and corrected the grammar.

Thank you.

Reviewer 2 Report

In the Review entitled “Non-coding RNAs and liquid biopsy in lung cancer: a literature review”, the authors present an overview of the current research landscape of blood-based diagnostic and prognostic tests to replace current standard-of-care tissue-based tests. While the authors do cover a wide range of current research efforts, there are several deficiencies detailed in the comments below.

Major Comments:

1. The description of miRNA biogenesis (lines 163-171) is not entirely correct and doesn’t mirror Figure 1, as exemplified by DGCR in the figure and Pasha in the text. Specifically, the primary miRNA transcript (pri-miRNA) can be transcribed from its own promotor either as part of a polycistron or single hairpin or arise from excised introns. The pri-miRNA is process into 70-100nt long pre-miRNA hairpins with a 3-prime overhang by the microprocessor complex (DGCR8/Drosha) in the nucleus. The pre-miRNA is exported to the nucleus (exportin-5) where it is processed by Dicer and the RISC complex into 18-23nt mature miRNA. Importantly, each pre-miRNA hairpin can produce 2 mature miRNA; one from each of the arms with -5p indicating the 5-prime arm and -3p indicating the 3-arm.

2. Relatedly, as depicted in Figure 1, it appears that hairpins for pri- and pre-miRNA forms are double stranded. These are single stranded RNA molecules that bond intramolecularly to form the secondary hairpin structure.

3. On line 172, the authors state that “…accurate miRNA expression is mainly determined by a regulated biogenesis process, which occurs at the transcriptional level of Drosha, Dicer and RISC, but also by its destabilization and degredation.” This is not correct. While regulation of these components does affect global miRNA expression, expression of each individual miRNA is regulated on the transcriptional level and transcription factors bound to the promotors or other DNA elements.

4. In Table 2 and the associated text, the authors identify miRNAs and targets with established functions in lung cancer organized by the hallmarks of cancer as described by Hanahan and Weinberg, originally proposed in 2000 and updated to 10 hallmarks in 2011. However, the authors omit 2 of these 10 hallmarks: Tumor-promoting Inflammation and Genome Instability & Mutation. Please comment.

Minor Comments:

1. The language is generally acceptable. However, many idioms and colloquialisms are used instead of specific language. This results in a lack of clarity. A scientific english copy-edit service is recommended.

2. It is unclear to the review why the authors focus the review on male lung cancer with the opening sentence of the introduction (line 32): “Lung cancer represents…in males on a global level;”

3. On line 123, the authors state “miRNAs represent such tissue”. MicroRNAs are a small RNA molecule, not a tissue.

Author Response

We are very grateful for all the suggestions and for carefully and thoroughly reading our paper. Hopefully, we have addressed all the comments and suggestions and have modified the text accordingly.

We added comments on the side of the manuscript where changes were suggested.

Point 1. The description of miRNA biogenesis (lines 163-171) is not entirely correct and doesn’t mirror Figure 1, as exemplified by DGCR in the figure and Pasha in the text. Specifically, the primary miRNA transcript (pri-miRNA) can be transcribed from its own promotor either as part of a polycistron or single hairpin or arise from excised introns. The pri-miRNA is process into 70-100nt long pre-miRNA hairpins with a 3-prime overhang by the microprocessor complex (DGCR8/Drosha) in the nucleus. The pre-miRNA is exported to the nucleus (exportin-5) where it is processed by Dicer and the RISC complex into 18-23nt mature miRNA. Importantly, each pre-miRNA hairpin can produce 2 mature miRNA; one from each of the arms with -5p indicating the 5-prime arm and -3p indicating the 3-arm.

 Response 1. We have clarified the DGCR/Pasha data and have added more explanations to the miRNA syntesis paragraph as suggested. We modified the image accordingly.

Point 2. Relatedly, as depicted in Figure 1, it appears that hairpins for pri- and pre-miRNA forms are double stranded. These are single stranded RNA molecules that bond intramolecularly to form the secondary hairpin structure.

Resonse 2. We tried to depinct the intramoleculary bond of RNA in the image and we tried to highlight in red the future mature miRNA strand. For a proper understanding, we now added the suggested explanation in the description below Figure 1.

Point 3. On line 172, the authors state that “…accurate miRNA expression is mainly determined by a regulated biogenesis process, which occurs at the transcriptional level of Drosha, Dicer and RISC, but also by its destabilization and degradation.” This is not correct. While regulation of these components does affect global miRNA expression, expression of each individual miRNA is regulated on the transcriptional level and transcription factors bound to the promotors or other DNA elements.

Response 3. We corrected as suggested.

Yang et al used the frase “To ensure correct miRNA expression, the miRNA biogenesis process is regulated at the transcriptional, DROSHA, DICER and RISC level.”  We misinterpreted and mistook by adding “mainly” to the paragraph.

In Table 2 and the associated text, the authors identify miRNAs and targets with established functions in lung cancer organized by the hallmarks of cancer as described by Hanahan and Weinberg, originally proposed in 2000 and updated to 10 hallmarks in 2011. However, the authors omit 2 of these 10 hallmarks: Tumor-promoting Inflammation and Genome Instability & Mutation. Please comment.

Response 4. Unfortunately we have omitted to add these 2 hallmarks, but we tried to remediate this.

We have now grouped the „avoiding the immune response” with „tumor-promoting inflammation” as they are both related to PD1/PDL-1. Tumor-promoting inflamation is achieved by PDL-1. When PD-1 cooperates with PD-L1, inhibitory signals are originated and prevent tumor cell annihilation. This is the data that we were able to find up to now.

In the last row of the table we have now also added miRNas associated with „Genome instability and mutation”.  [67],[68]

Minor comments

The language is generally acceptable. However, many idioms and colloquialisms are used instead of specific language. This results in a lack of clarity. A scientific english copy-edit service is recommended.

Response 1: We have also used for the first manuscript a scientific English edit service, but we have revised the manuscript again with another service, as suggested. Hopefully, it is now clearer and more accurate.

It is unclear to the review why the authors focus the review on male lung cancer with the opening sentence of the introduction (line 32): “Lung cancer represents…in males on a global level;”

Response 2: We now added a sentence to cover the statistics for female population too.

On line 123, the authors state “miRNAs represent such tissue”. MicroRNAs are a small RNA molecule, not a tissue. We corrected this, thank you .

Reviewer 3 Report

The manuscript by HaranguÈ™ covers an interesting topic but is not clearly focused and its organization must also improve in order to offer an interesting and clear perspective to readers. The most remarkable observation is that there is a mixing throughout the text about results on circulating DNA, whose results have entered clinical validation, and circulating RNA, whose results are in a earlier phase of development.  In fact, some paragraphs, as for example “Detecting EGFR, ALK and ROS1 in plasma” deals with topics already treated in a previous paragraphs” Liquid biopsy in lung cancer”), but repetitive content and confused text organization should be avoided. In addition, in the Abstract, the last sentence introduces the content of the review, but this content “carcinogenesis in the lung and lung cancer genomics. It also addresses present and future possible roles of liquid biopsy in this pathology.” appears not coherent with the title. The aim of the manuscript should be focused better, as for example in the last lines of the Introduction, which are more coherent with the title. Besides, it is not easy why the paragraphs introducing a role for miRNA was inserted at the end of a paragraph trying to describe the main mutagenic events associated to lung cancer development. As a matter of fact, the paragraph describing the genetic mutation that have been up to now associated to lung cancer should be not exhaustive but clearer (what is the percentage of lung cancer associated to mutation? What is the percentage of cancer not associated to mutation? Some fundamental information should be reported and specific references to reviews on the topic added) and there is no point in introducing non-coding RNA in this paragraph. Some sentences appear isolated in the context and not integrated with the rest of the text, as for example line 370 “Until recently, the majority of the research has examined ncRNAs from patient tissues obtained by invasive methods.”

Other points

Line 33 “Due to late diagnosis, together with the lack of curative therapies, in an advanced disease stage,…” Comma should be eliminated

Line 48 In the sentence “…new mutations, structural variations [8], number of reads [9], microRNAs [10] and analyse DNA-methylation [11]…” it is not clear what do authors mean for  number of reads

Line 53 The sentence “Therapy could also be target epigenetic changes, reprogram tumor cells and make them more vulnerable to succeeding treatments.” appears not correct

Lines 92-94 The sentence “The original cells for cancers of the lung have been proven to be involved in airway repair processes. The initiating cells of NSCLC are the alveolar type II epithelial cells for Kirsten rat sarcoma 2 viral oncogene homolog (KRAS)-driven adenocarcinomas, while for squamous cell carcinoma, the exact origin is still unclear [21].” is a little bit confusing and should be improved, because in Table 1 it is indicated that only 32.2% adenocarcinomas are associated with KRAS Mt 32.2%.

Sentences should be clearly explained and referenced, as for example at line 151 “…sometimes even more than protein coding genes,…”, otherwise they are confusing and possibly incorrect. Original studies demonstrating this evidence should be listed and discussed.

Line 237-241 the paragraph is confused and should be carefully integrated with the text below, in particular it is not possible to get a clear picture of the role of HOTAIR

Line 324 In referring to extracellular vesicles, authors should introduce the concept of oncosomes

Author Response

We are very grateful for all the suggestions and for carefully and thoroughly reading our paper. Hopefully, we have addressed all the comments and suggestions and have modified the text accordingly.

We added comments on the side of the manuscript where changes were suggested.

The manuscript by HaranguÈ™ covers an interesting topic but is not clearly focused and its organization must also improve in order to offer an interesting and clear perspective to readers. The most remarkable observation is that there is a mixing throughout the text about results on circulating DNA, whose results have entered clinical validation, and circulating RNA, whose results are in a earlier phase of development.  In fact, some paragraphs, as for example “Detecting EGFR, ALK and ROS1 in plasma” deals with topics already treated in a previous paragraphs” Liquid biopsy in lung cancer”), but repetitive content and confused text organization should be avoided.

Response: The manuscript tries to cover different research and data concerning liquid biopsy in lung cancer, and we felt the need to cover both data about mutation such as EGFR, ALK, ROS1, with more interest for clinicians, but also data concerning cDNA and cRNA. As suggested, we have moved some paragraphs that were in the chapter about liquid biopsy in lung cancer, in the chapter concerning EGFR, ALK and ROS1 for a better organization of the text.

In addition, in the Abstract, the last sentence introduces the content of the review, but this content “carcinogenesis in the lung and lung cancer genomics. It also addresses present and future possible roles of liquid biopsy in this pathology.” appears not coherent with the title. The aim of the manuscript should be focused better, as for example in the last lines of the Introduction, which are more coherent with the title.

Response: We have changed the last sentence of the abstract and tried to be more coherent with the title.

This article addresses present and future possible roles of liquid biopsy in lung cancer. It also discusses how the complex role of noncoding RNAs in lung tumorigenesis could influence the management of this pathology.

 Besides, it is not easy why the paragraphs introducing a role for miRNA was inserted at the end of a paragraph trying to describe the main mutagenic events associated to lung cancer development. As a matter of fact, the paragraph describing the genetic mutation that have been up to now associated to lung cancer should be not exhaustive but clearer (what is the percentage of lung cancer associated to mutation? What is the percentage of cancer not associated to mutation? Some fundamental information should be reported and specific references to reviews on the topic added) and there is no point in introducing non-coding RNA in this paragraph.

Response: We inserted the last paragraph in order to provide a connection with the next chapter. We have taken into consideration your suggestion and removed it.

Table 1 shows selected common somatic cancer aberrations and affected genes in lung cancer with percentages.

Some sentences appear isolated in the context and not integrated with the rest of the text, as for example line 370 “Until recently, the majority of the research has examined ncRNAs from patient tissues obtained by invasive methods.”

Response: We added a new sentence in order to better introduce the chapter. (see comment in manuscript).

Other points

Line 33 “Due to late diagnosis, together with the lack of curative therapies, in an advanced disease stage,…” Comma should be eliminated

Response: corrected

Line 48 In the sentence “…new mutations, structural variations [8], number of reads [9], microRNAs [10] and analyse DNA-methylation [11]…” it is not clear what do authors mean for  number of reads

Response: We modified this paragraph in order to be clearer.

High-throughput sequencing can generate large amounts of data concerning the lung cancer exome and genome in order to discover new mutations, structural variations [8], microRNAs [10] and analyse DNA-methylation [11]. Deep sequencing methods aim for high number of unique reads in every region of a sequence, therefore raising the sequencing accuracy [9].

Line 53 The sentence “Therapy could also be target epigenetic changes, reprogram tumor cells and make them more vulnerable to succeeding treatments.” appears not correct

Response: We modified this sentence as following:

Future therapies might also target epigenetic changes, reprogram tumor cells by making them more vulnerable

Lines 92-94 The sentence “The original cells for cancers of the lung have been proven to be involved in airway repair processes. The initiating cells of NSCLC are the alveolar type II epithelial cells for Kirsten rat sarcoma 2 viral oncogene homolog (KRAS)-driven adenocarcinomas, while for squamous cell carcinoma, the exact origin is still unclear [21].” is a little bit confusing and should be improved, because in Table 1 it is indicated that only 32.2% adenocarcinomas are associated with KRAS Mt 32.2%.

Response: We tried to improve this paragraph. Also, we added more recent data about the origin of squamous cell carcinoma.

The original cells for cancers of the lung are known to be involved in airway repair processes. For Kirsten rat sarcoma 2 viral oncogene homolog (KRAS)-driven adenocarcinomas, the most common type of lung adenocarcinomas, the initiating cells were found to be the alveolar type II epithelial cells [21].  Squamous cell carcinoma most often origins from the bronchial epithelium of central, larger bronchi and has morphological features such as intercellular bridges, squamous pearl formation and individual cell keratinization [22].

Sentences should be clearly explained and referenced, as for example at line 151 “…sometimes even more than protein coding genes,…”, otherwise they are confusing and possibly incorrect. Original studies demonstrating this evidence should be listed and discussed.

Response: We have deleted this part, in order to be concise and correct.

The ncRNAs can influence lung tumorigenesis and proliferation and have therefore attracted attention as future possible biomarkers or therapeutic targets [31].

Line 237-241 the paragraph is confused and should be carefully integrated with the text below, in particular it is not possible to get a clear picture of the role of HOTAIR

Response:  We added a sentence to make a better connection to the paragraph below and removed some details that were not clear.

Also we changed the order of the paragraphs in order to provide a better picture of the role of HOTAIR (see comments added at the manuscript).

Line 324 In referring to extracellular vesicles, authors should introduce the concept of oncosome

Response: Thank you for this suggestion, we have introduced the concept as follows:

…One example are the oncosomes, large plasma membrane-derived EVs spontaneously released by tumor cells that present metalloproteinases with invasive properties

Round 2

Reviewer 2 Report

The reviewer commends the authors for their responsiveness to the comments. However, the reviewer is still concerned with the depiction of the RNA hairpin secondary structure as a dsRNA stem-loop.

Major Comments:

Point 1. Sufficiently addressed. The revised text is now factually correct.

Point 2. Not sufficiently addressed. In Figure 1, it still appears that the hairpins are double stranded. These are single stranded RNA molecules that bond intramolecularly to form the secondary hairpin structure.

Additionally, in the emended text, the authors description mirrors the depiction of each RNA hairpin as a stem-loop of double-stranded RNA with each arm generating a miRNA:miRNA duplex. In fact, the formation of the miRNA hairpin secondary structure allows for segments of dsRNA that RNase III enzymes can recognize.

3. Sufficiently addressed.

4. Sufficiently addressed.

Minor Comments:

1. Sufficiently addressed.

2. Sufficiently addressed.

3. Sufficiently addressed.

Author Response

Response to Reviewer 2 Comments

Point 2. Not sufficiently addressed. In Figure 1, it still appears that the hairpins are double stranded. These are single stranded RNA molecules that bond intramolecularly to form the secondary hairpin structure.

We have modified the hairpins according to your suggestions.

Additionally, in the emended text, the authors description mirrors the depiction of each RNA hairpin as a stem-loop of double-stranded RNA with each arm generating a miRNA:miRNA duplex. In fact, the formation of the miRNA hairpin secondary structure allows for segments of dsRNA that RNase III enzymes can recognize.

We corrected in the text as suggested, hopefully it is clearer now.

"Pri-miRNAs can be transcribed from its own promotor from intronic, intergenic or polycistronic loci or as part of a single hairpin. The precursors are first transcribed by RNA polymerase II and then intranuclearly processed by Drosha, an RNase III enzyme, and Pasha, a double-stranded RNA (dsRNA)-binding protein to form the pre-miRNA. This product is transported to the cytoplasm, where it is cleaved by Dicer, another RNase III enzyme. The endoribonuclease removes the loop that joins the 3’ and 5’ arm and forms the miRNA: miRNA duplex. The duplex unwinds and one strand assembles into RNA-induced silencing complex (RISC), where it plays the role of a template that identifies the complementary mRNA and downregulates its expression by either direct degradation or translational repression. The other strand of the duplex is degraded."

"Canonical miRNA genesis and processing pathway. The miRNA gene is transcribed in the nucleus from intronic, intergenic or polycistronic loci by RNA polymerase II or III and forms a transcript that is called primary miRNA (Pri-miRNA). Pri-miRNA hairpins are double-stranded RNA (dsRNA) structures cleaved by Drosha, an RNAse III type enzyme, and Pasha, also known as DiGeorge syndrome critical region gene 8 (DGCR8), and form a 70-100 nucleotide long precursor (pre-miRNA).  Pre-miRNA hairpin is transported to cytoplasm by the exportin-5 and RanGTP cofactor and then processed by the Dicer complex, another RNase III enzyme, into a miRNA duplex. The unwinding of the duplex forms the 18-23nt mature miRNA. One strand of the duplex binds to Ago and forms RNA-induced silencing complex (RISC). Once loaded, the RISC can target and induce mRNA negative expression by cleavage, translational inhibition or destabilization and degradation."

Reviewer 3 Report

The revised version of the manuscript by HaranguÈ™ et al has improved its organization and clarity. Some sentences need to be fixed:

Line 107 Please separate Table 1 legend from the rest of the text Please check characters dimension in the following sentences “demonstrated, using gene expression microarray analysis, that activation of the NF-kB subunit 2 (p52) is a mediator in the development of lung cancer, as it promotes proliferation, and could also be taken into consideration as a new therapeutic target in the future [26].” The sentence “where it is processed by Dicer, a different RNase III enzyme, to form the miRNA: miRNA duplex.” and “One chain of the duplex generally binds to the RNA-induced silencing complex (RISC)” have to be checked for grammar/language use Line 333 Correct number The definition of oncosomes, microvesicles, exosomes, ectosomes and lipoproteins as “phospholipid-protein complexes” is not correct. Please reformulate the definition. Please check the sentence “One example are the oncosomes”, perhaps authors should mean “As for example”

Author Response

Response to Reviewer 3 Comments

Line 107 Please separate Table 1 legend from the rest of the text

Modified

Please check characters dimension in the following sentences “demonstrated, using gene expression microarray analysis, that activation of the NF-kB subunit 2 (p52) is a mediator in the development of lung cancer, as it promotes proliferation, and could also be taken into consideration as a new therapeutic target in the future [26].”

Modified

The sentence “where it is processed by Dicer, a different RNase III enzyme, to form the miRNA: miRNA duplex.” and “One chain of the duplex generally binds to the RNA-induced silencing complex (RISC)” have to be checked for grammar/language use

We asked the help of the University translator and changed according to their suggestion, hopefully it is what you suggested.

This product is transported to the cytoplasm, where it is processed by Dicer, another RNase III enzyme, to form the miRNA: miRNA duplex.

The miRNA: miRNA unwinds and the mature miRNA assembles into RNA-induced silencing complex (RISC),

Line 333 Correct number

Corrected

The definition of oncosomes, microvesicles, exosomes, ectosomes and lipoproteins as “phospholipid-protein complexes” is not correct. Please reformulate the definition.

We reformulated: “Cells exchange different vesicles such as oncosomes, microvesicles, exosomes, ectosomes and lipoproteins.”

Please check the sentence “One example are the oncosomes”, perhaps authors should mean “As for example”

Corrected: “As for example, the oncosomes are large plasma membrane-derived EVs spontaneously released by tumor cells that present metalloproteinases with invasive properties.”